# An Accelerated-Based Evaluation Method for Corrosion Lifetime of Materials Considering High-Temperature Oxidation Corrosion

**Hongbin Zhang** [1,2]**, Shuqiang Liu** [2]**, Peibo Liang** [2,3,*]**, Zhipeng Ye** [2,3] **and Yaqiu Li** [2,4,*]

1　School of Materials Science and Engineering, South China University of Technology, Guangzhou 510641, China
2　China Electronic Product Reliability and Environmental Testing Research Institute, Guangzhou 511370, China
3　Guangdong Provincial Key Laboratory of Electronic Information Products Reliability Technology, Guangzhou 511370, China
4　Key Laboratory of Active Medical Devices Quality & Reliability Management and Assessment, Guangzhou 511370, China
*　Correspondence: liang_pb@163.com (P.L.); ermao13@163.com (Y.L.)

**Abstract:** In the realm of industrial automation, corrosion represents one of the primary modes of failure, especially in the case of armored thermocouples exposed to temperatures ranging between 1073.15–1373.15 K. In this context, the selection of metal materials that can withstand high-temperature oxidation and corrosion is of paramount importance. Typically, the corrosion resistance of a given metal material is assessed by measuring the "annual corrosion rate" or "corrosion depth", which can provide an estimated life expectancy value. However, such an approach fails to account for the individual characteristics of the material, and thus does not conform to objective laws. Rather, the corrosion life of a batch of metallic materials should follow the Weibull distribution, or possibly a normal distribution, as per recent studies that have examined the high-temperature oxidation corrosion mechanism of machine or core components. This investigation effectively combines the standard approach for evaluating metal corrosion resistance in the field of materials with the method of assessing component life in the domain of reliability. Furthermore, we consider the individual differences among materials and provide the life distribution function of metals in corrosive environments and thereby refine the evaluation of metal corrosion resistance. This study ultimately establishes a thermocouple accelerated life evaluation model that enhances the accuracy and efficiency of life evaluations for related products.

**Keywords:** armored thermocouples; high-temperature oxidation corrosion; oxidation behavior; metal material lifetime distribution; accelerated life testing

## 1. Introduction

Thermocouples, as typical industrial automation instruments, are widely used in various industrial processes, such as metallurgy, the chemical industry, and power generation, and serve as essential temperature control units in industrial production lines. Their significance in industrial production cannot be overstated. To illustrate, in the ethylene cracking process, the performance of the cracking furnace directly impacts the economic efficiency of the entire system. The exit temperature of the cracking furnace serves as a critical indicator of the cracking furnace, which affects not only the determination of the operation status of the furnace but also the yield of ethylene. Therefore, it is crucial to ensure that the temperature instrumentation maintains a precise and effective temperature measurement over a prolonged period. Given the high temperature and fast flow rate of the cracking gas in the cracking furnace, the temperature instrument typically employs an inserted armored thermocouple, as depicted in Figure 1. The protective casing of the

thermocouple is composed of an alloy with excellent high-temperature and wear resistance properties to ensure the temperature instrument's high accuracy and long working life.

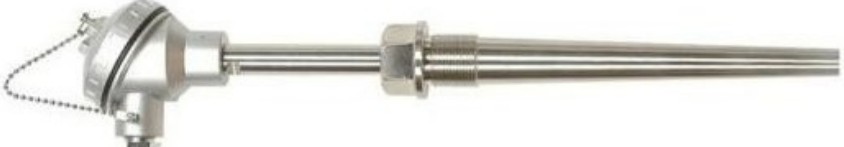

**Figure 1.** Typical industrial armored thermocouples.

Armored thermocouples are widely utilized due to their high accuracy, reliability, and durability [1]. However, their operation in complex environments presents a challenge when it comes to monitoring their reliability and predicting their remaining lifespan. Accelerated reliability testing is an effective technique used to address this issue, wherein test units are subjected to severe conditions such as higher temperatures, voltages, and pressures to induce accelerating stresses [2] and thereby saves time and costs. Through data analysis, the product lifetime distribution is estimated so as to predict its reliability. The Arrhenius model is the classic evaluation model for accelerated testing. Notably, a prerequisite for the success of an accelerated test evaluation is that the failure mechanism does not change under more severe stress conditions. Moreover, several researchers have studied the failure mechanism change point for the accelerated model [3,4]. Due to the highly reliable and more complex failure mechanisms, lifetime date collection become technically challenging. The accelerated degradation model is developed for the purpose of obtaining more measurement data with fewer samples. The framework of the degradation model typically comprises both the general path model and the stochastic process model. The general path model uses the regression method to fit the degradation path function, and the stochastic process model such as the Wiener, Gamma, and Inverse Gaussian models [5] are available to consider the variability among the units. At present, accelerated testing is extensively applied to electronic devices and materials, including DC film capacitors [6] and IGBTs [7], as well as rubber [8] and equipment products such as lithium batteries [9], turbochargers [10], etc. The accuracy of the reliability assessment or remaining life prediction of these products is effectively improved by accelerated testing.

The failure of furnace temperature instrumentation due to the oxidation corrosion of thermocouple protection sleeves in high-temperature environments is a well-known issue. In such cases, the failure of the protective casing can lead to severe damage to the internal thermocouple temperature measurement system and complete instrument failure as a result of the scouring effect of the high flow rate media. The failure of thermocouples in crucial components can lead to substantial economic losses that are difficult to rectify. Given the challenging working environment of thermocouples, the selection of protective tube materials in the current market is highly meticulous. Different materials are chosen for different working conditions, including copper, stainless steel, nickel-based high-temperature alloys, iron-based corrosion-resistant alloys, and precious metals, among others [11–17].

Thus, the selection of high-temperature oxidation corrosion-resistant metal materials and the evaluation of the corrosion resistance characteristics of specific metals have become crucial elements in the development of thermocouples. As new designs for thermocouple protection tubes that can withstand corrosion are developed, the service life of these tubes is extended significantly and usually measured in "years". Accelerated testing is used to evaluate the reliability of the thermocouples due to its successful application to other product assessments with limited time and costs. Therefore, based on the well-established corrosion mechanism, this study primarily concentrates on substantiating the applicability of the accelerated reliability test on armored thermocouple products and analyzing the advantages compared to the conventional method.

## 2. Mechanism and Model

Thermocouples utilized in ethylene cracking furnaces operate within an oxygen-rich environment at 1073.15–1123.15 K. As such, the principal mode of failure for thermocouple protection tubes is high-temperature oxidation corrosion.

### 2.1. The High-Temperature Oxidation Corrosion Mechanism

In Figure 2, it is demonstrated that the free energy of the oxide formation of typical structural material components is negative at service temperature. This finding suggests that the reaction between these materials and oxygen, which results in the formation of oxides, is thermodynamically spontaneous. Despite this, the materials do not instantaneously transform into oxides when exposed to air due to the presence of a complete oxide layer that functions as an insulator, which allows the oxides to shield the metal surface from oxygen and thereby avoid the rapid oxidation of the material.

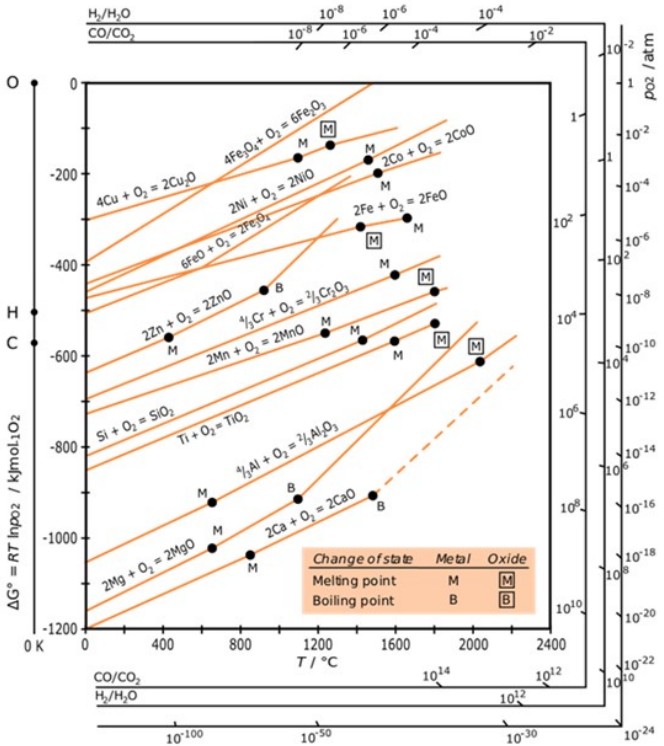

**Figure 2.** The Ellingham diagram that represents the standard free energy of oxide formation for common structural materials [18].

Figure 3 schematically represents the oxidation process of metals in high-temperature air, as observed through various studies [19–23]. The initial step involves the rapid growth of oxide nucleation, which is also a swift process at room temperature. Once a complete oxide film forms, oxygen gains electrons at the gas/oxide film interface, while metal loses electrons at the oxide film/metal interface. This process leads to the formation of a concentration gradient of anions, cations, and electrons at the oxide film, which diffuse through the oxide film in opposite directions to complete the entire oxidation process. Multiple studies have demonstrated that the oxides of typical structural material elements behave as p-type semiconductors, with cation vacancies serving as the primary carriers of conductivity. Consequently, the oxidation rate of these materials is generally governed by the diffusion rate of cations in the oxide film.

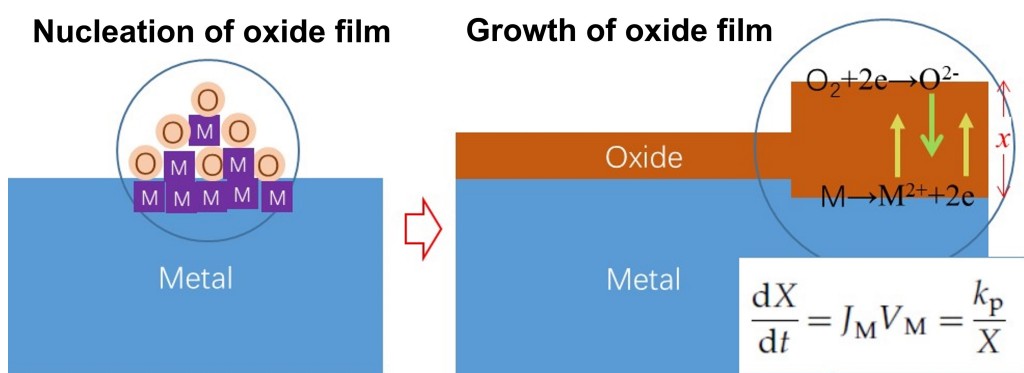

**Figure 3.** Schematic diagram of nucleation and growth of oxide film on metal surface.

Both thermodynamic theoretical analyses and experimental studies indicate that dense corrosion products can be described by a parabolic trend in high-temperature oxidation dynamics (corrosion volume versus time) as follows:

$$(\Delta w)^2 = k_p t \tag{1}$$

in which $\Delta w$ denotes the weight gain per unit surface area and $k_p$ and $t$ represent the corrosion rate and time, respectively.

Wagner's theory posits that the parabolic law indicates a strict dependence of the corrosion rate on the diffusion of the group elements present in corrosion as they move through the corrosion product layer. Additionally, the corrosion rate is inversely proportional to the thickness of the corrosion product layer. The Arrhenius relations are available to express the parameters associated with the diffusion-controlled process and its temperature dependence.

$$k_p(T) = k_0 \exp(-E_a/k_B T) \Leftrightarrow \ln(k_p) = \ln(A) - (E_a/k_B T) \tag{2}$$

in which $E_a$ denotes the activation energy and $k_B$ and $T$ denote the Boltzmann's constant and temperature, respectively.

Equation (1) establishes a quantitative model of high-temperature oxidation corrosion by determining the mass of oxide $\Delta\omega$ generated by a metal at various temperature steps within a given time. Typically, in material corrosion studies, "corrosion rate per year" and "corrosion depth" are used to illustrate the corrosion resistance of metal materials [24,25]. Two or three parallel samples are produced at the same temperature step, and their weight gain data are collected to calculate the corrosion rate $k_p$ of the metal at different temperatures, which can then be used to determine its lifetime value (a definite value) at that temperature if a threshold value is provided. However, it is crucial to acknowledge that both the corrosion rate and lifetime value are approximate mathematical values and that small differences exist between parallel samples due to factors such as surface roughness and processing. From a natural science perspective, each sample represents a unique individual, and a sufficient number of samples undergoing high-temperature oxidation corrosion should follow either the Weibull or normal distribution concerning the corrosion amount, corrosion rate, or life. Therefore, we propose the utilization of the pseudo failure life analysis method for metal corrosion data, which conceptualizes the corrosion process as a degradation process of metal properties or life. We consider different temperature gradients and parallel samples as independent entities, analyze the corrosion rate and life of each parallel sample separately, and ultimately construct a pseudo failure life model of metal material corrosion.

### 2.2. Pseudo Failure Lifetime Model

Traditional life assessments and predictions primarily rely on theories and methods founded on mathematical statistics and life tests, with the statistical analysis primarily

focused on the life data, also known as the "Time to failure". However, obtaining sufficient failure data, or even any failure data, for highly reliable and long-life products within the limited test time presents significant challenges, which renders traditional life assessment methods insufficient.

The life consumption process of highly reliable long-life products can typically be attributed to potential performance degradation resulting from failure mechanisms, which eventually lead to product failure. If the failure mechanism and degradation magnitude of a product are understood, the product's reliability can be determined by measuring the time when its performance reaches a critical level of degradation. This approach enables the extrapolation of the reliability of highly reliable long-life products by estimating their degradation pattern under a given stress, even if the actual failure time of the product is not observed. To this end, we propose the use of product performance degradation data to estimate the reliability and lifetime of highly reliable long-life products. This approach involves monitoring the performance state of highly reliable long-life products by using product performance degradation data, which can be employed to model and analyze the product life expectancy.

The slow degradation rate of metal materials during high-temperature oxidation results in an extremely small change in the amount of degradation over a long test time, which renders it challenging to obtain the product life characteristics through testing. In the case of metal materials, the degradation process equates to the corrosion process, with the amount of corrosion serving as the amount of performance degradation. To determine the characteristic life of metal materials, one can extrapolate the corrosion depth at a specific temperature based on the corrosion rate and estimate the time required to reach a critical level. This method provides an estimate of the characteristic life of the metal material.

Equation (2) characterizes the high-temperature oxidation corrosion process of metallic materials. Within the realm of reliability, the corrosion rate $k_p$ represents the degradation rate of the metal in this environment. The weight gain $d$ indicates the amount of degradation. Therefore, we can conclude that:

$$\frac{dM}{dt} = Ae^{-E_a/k_B T} \Rightarrow dM = Ae^{-E_a/k_B T}\, dt \tag{3}$$

where $M$ indicates the mass of the sample.

At time $t = t_0$, the metal is in its initial state, with a mass of $M_0$. When $t = t_1$, the mass of the metal changes to $M_1$. Assuming that temperature is independent of time, we have

$$\int_{M_0}^{M_1} dM = \int_{t_0}^{t_1} Ae^{-E_a/k_B T} dt \Leftrightarrow M_1 - M_0 = Ae^{-E_a/k_B T}(t_1 - t_0) \tag{4}$$

let $t = t_1 - t_0$ and we have

$$t = \frac{M_1 - M_0}{A} e^{E_a/kT} = Be^{E_a/k_B T} \Leftrightarrow \ln(t) = \ln(B) + (E_a/k_B T) \tag{5}$$

in which $B$ can be seen as the model parameters. If the metal fails at $t = t_1$, then $t$ equals the characteristic lifetime $\eta$ of the metallic material at a particular temperature. Therefore, the $\eta$ of a single sample can be expressed as

$$\eta = B\exp[Ea/(k_B T)] \Leftrightarrow \eta(T) = B\exp(C/T) \tag{6}$$

where C is also a model parameter and equals $Ea/k_B$.

### 2.3. Acceleration Factor Model Based on Weibull Distribution

Assuming that the product lifetime obeys the two-parameter Weibull distribution, the expression for its cumulative failure probability function $F(t)$ can be established as follows:

$$F(t) = 1 - e^{-(t/\eta)^{\beta}} \tag{7}$$

in which $\beta$ is the shape parameter of the Weibull distribution. And we have

$$\ln(-\ln(1 - F(t))) = \beta ln\left(\frac{t}{\eta}\right) = \beta ln(t) - \beta ln(\eta) \tag{8}$$

Performing a linear fit on the above expression for the same temperature enables us to obtain the Weibull distribution parameters $(\beta, \eta)$ of lifetime and probability at different temperatures, which can be used to analyze the reliability level of the product.

Let $t_0(F_0)$ denote the time required for the test to reach the cumulative failure (risk) probability $F_0$ and $t_1(F_0)$ denote the time required for the accelerated life test to reach the same cumulative failure (risk) probability under the other stress condition. Then, the acceleration factor can be derived as

$$AF = \frac{t_0(F_0)}{t_1(F_1)} = \exp\left[\frac{E_a}{k_B}\left(\frac{1}{T_0} - \frac{1}{T_1}\right)\right] \tag{9}$$

## 3. Experiments and Results

Consider the example of the thermocouple used in ethylene cracking furnaces, which operates in an environment ranging from 1073.15 to 1373.15 K, with the main failure mode being the high-temperature oxidation corrosion of the protection jacket, as shown in Figure 4. Since high-temperature oxidation corrosion results in uniform corrosion, the metal material undergoes relatively uniform thinning throughout the corrosion process and eventually loses its protective function for the thermocouple. Therefore, the failure process of the thermocouple is primarily attributed to the thickness of the metal material, which undergoes corrosion-induced performance degradation. This phenomenon aligns with corrosion theory and the pseudo failure degradation model proposed in the previous section.

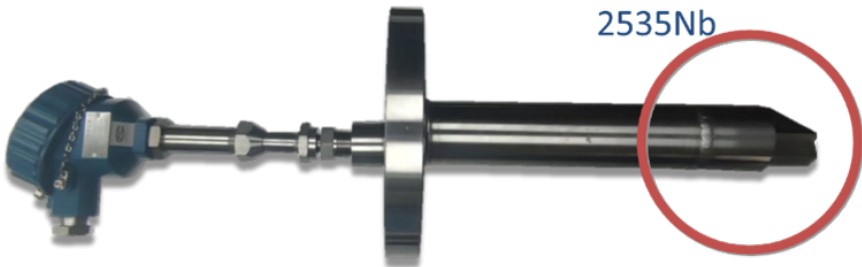

**Figure 4.** Thermocouple sample used in ethylene cracking furnaces.

### 3.1. Consistency Test of Mechanism

The portion of the armored thermocouple that comes in direct contact with the high-temperature medium is made of a 2535 *Nb* alloy, which is an iron–nickel-based heat-resistant alloy. This alloy can be fabricated into welding wire, welding rods, and other products and is primarily used in the petrochemical industry for centrifugal casting tube reformer welding, with a working temperature range of 1073.15 to 1373.15 K. The chemical composition of this alloy is presented in the following Table 1.

**Table 1.** Chemical composition of the 2535 *Nb* alloy samples.

| Alloy Compositions (%) | | *Ni* | *Cr* | *Nb* | *C* | *Mn* | *Si* | *P* | *S* | *Fe* |
|---|---|---|---|---|---|---|---|---|---|---|
| 2535 *Nb* | min | 32 | 24 | 0.7 | 0.35 | 0 | 0 | 0 | 0 | Bal. |
| | max | 37 | 27 | 1.5 | 0.45 | 2.0 | 0.04 | 0.04 | 0.04 | |

Given the superior creep endurance, persistent strength, and robust resistance to high-temperature gas corrosion offered by the 2535 *Nb* alloy, a preliminary analysis of its oxidation resistance properties was conducted. This analysis was performed for a

sample (1273.15 K, 40 h), the microscopic morphology of which is illustrated in Figure 5. The composition analysis is detailed in Table 2.

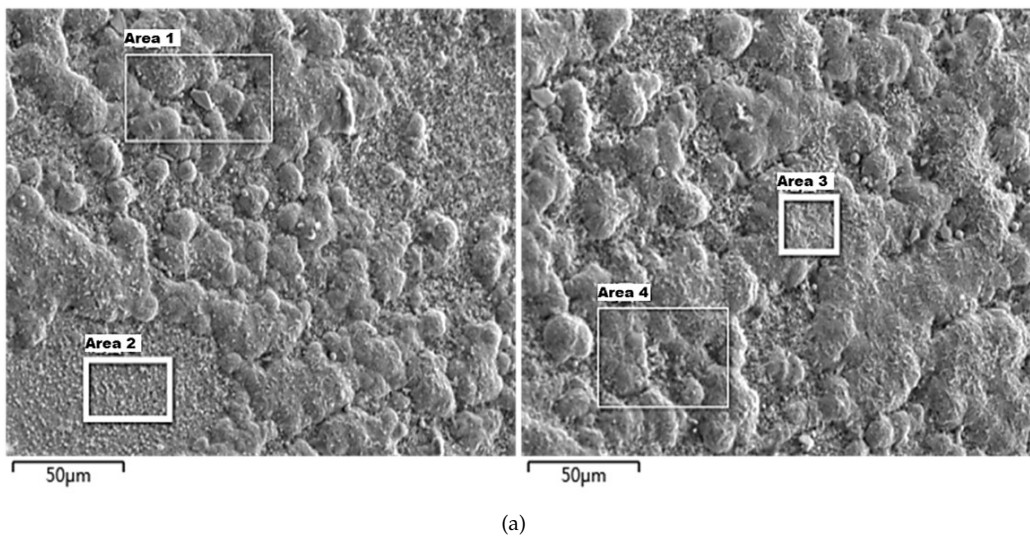

(a)

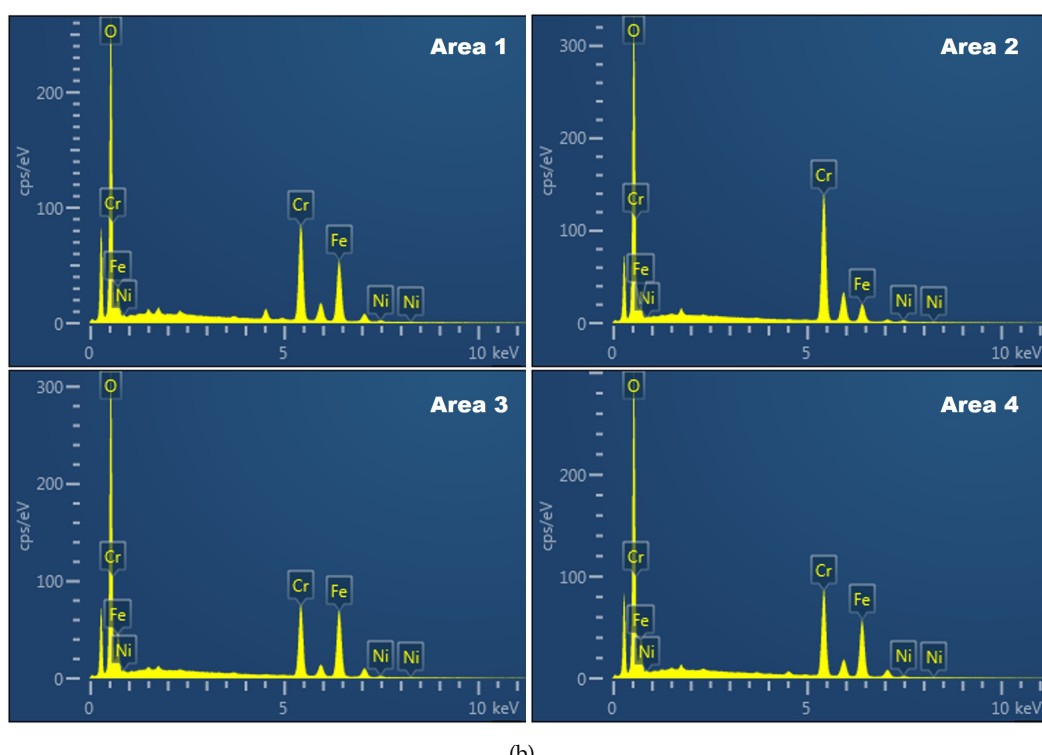

(b)

**Figure 5.** (**a**) Microscopic morphology and (**b**) EDS results of the sample surface.

**Table 2.** Analysis of sample surface composition (wt.%).

| Element | Area 1 | Area 2 | Area 3 | Area 4 |
|---------|--------|--------|--------|--------|
| O  | 24.7  | 27.16 | 26.71 | 26.5  |
| Cr | 36.06 | 58.44 | 27.77 | 34.64 |
| Fe | 37.85 | 12.69 | 44.16 | 37.67 |
| Ni | 1.4   | 1.72  | 1.36  | 1.19  |

As presented in Figure 6, the XRD analysis results indicated that the metal surface layer predominantly generated two types of oxides, $Cr_2O_3$ and $Ni (Cr_2O_4)$. The formation of this oxide film could effectively mitigate the corrosion rate.

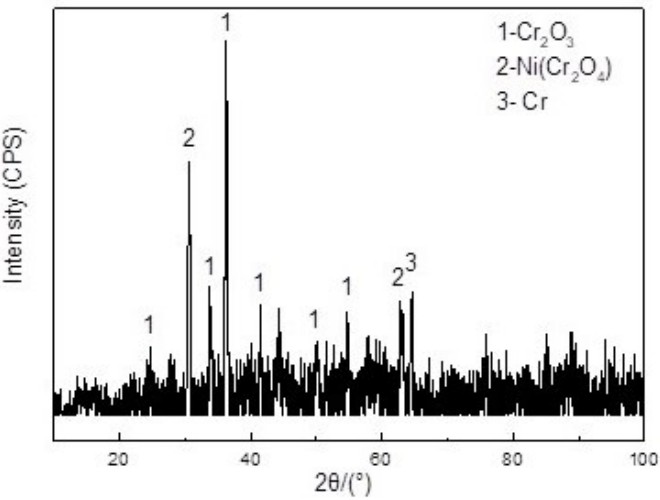

**Figure 6.** The XRD results of sample surface.

To conduct the mechanistic consistency tests, we selected three stations to analyze the instruments at 1163.15 K, 1233.15 K, and 1323.15 K (in air). In order to clarify the oxidation corrosion process of the 2535 *Nb* alloy at a specific temperature, samples were analyzed after 100 h, 300 h, and 500 h of operation. The macroscopic appearance of the alloy sample after oxidation is presented in Figure 7.

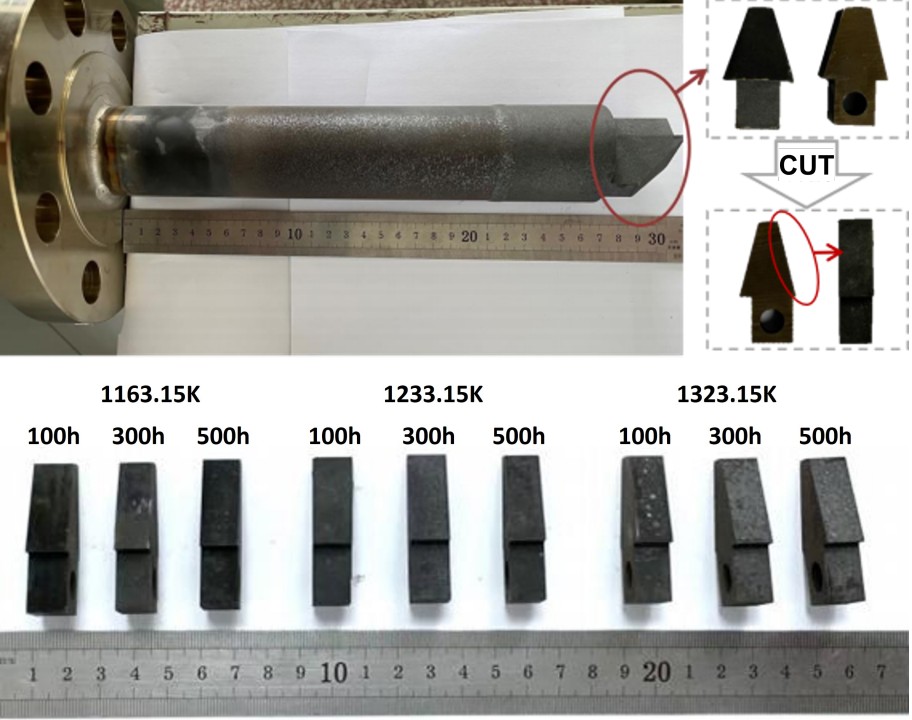

**Figure 7.** The cut samples after high-temperature tests.

### 3.1.1. The Oxidation Morphology of 2535 *Nb* Alloy Sample at 1163.15 K

The microscopic morphology of the 2535 *Nb* alloy sample after oxidation at 1163.15 K is presented in Figure 8. The level of oxidation corrosion at this temperature was relatively

mild, as visible streaks from processing and polishing the sample were still present on the surface. Upon oxidation for 100 h, the alloy surface displayed a continuous and dense oxide film that was primarily composed of $Cr_2O_3$ with a small quantity of $Mn$-$Cr$ spinel distributed on the surface layer. Additionally, some oxide nodules, identified as $(Cr, Fe)_2O_3$ oxides, were observable on the surface of the oxide film. After 300 and 500 h of oxidation, the oxide film demonstrated local flaking and self-healing, and eventually formed a new oxide film membrane. The $Cr_2O_3$ film continued to grow throughout the oxidation process, with the thickness of the oxide film being approximately 2, 3, and 4 μm after 100, 300, and 500 h of oxidation, respectively, which demonstrates the outstanding oxidation resistance of the alloy at 1163.15 K.

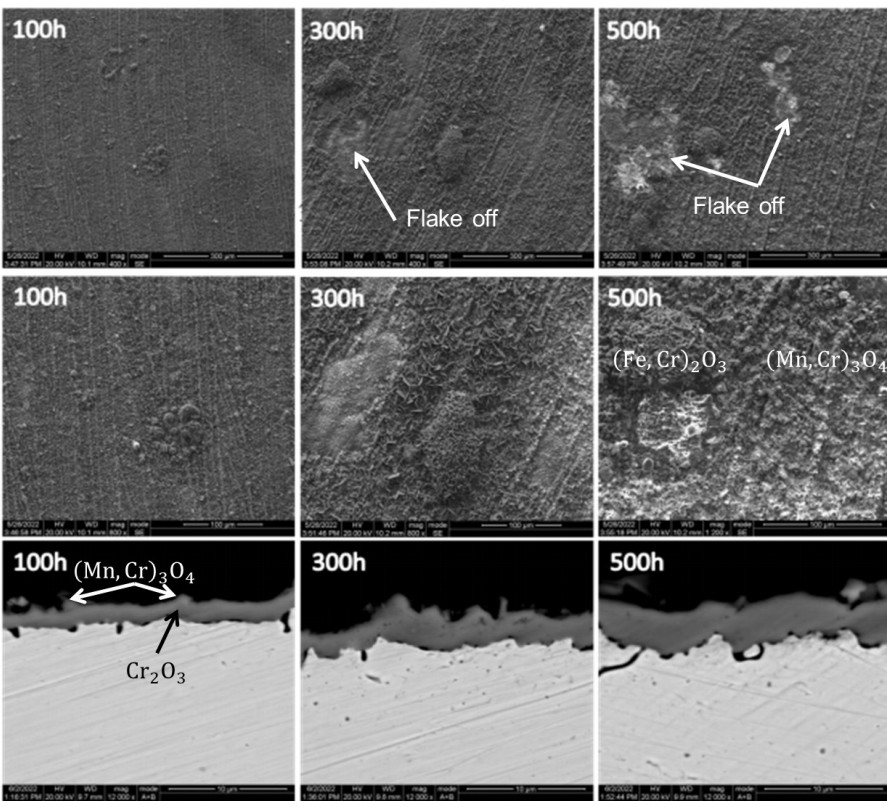

**Figure 8.** The oxidation morphology of 2535 $Nb$ alloy sample at 1163.15 K after 100 h, 300 h, and 500 h.

### 3.1.2. The Oxidation Morphology of 2535 $Nb$ Alloy Sample at 1233.15 K

Figure 9 depicts the microscopic morphology of the 2535 $Nb$ alloy sample subsequent to oxidation at 1233.15 K. Comparative analysis with the oxidation outcomes of the alloy at 1163.15 K indicated that the corrosion of the alloy was escalated upon oxidation at 1233.15 K. The alloy's surface morphology and alterations were consistent with those observed at 1163.15 K oxidation, albeit with coarsened oxide particles. The coarsening of the oxide particles signifies a significant increase in the $Mn$-$Cr$ spinel content developed in the most superficial layer, while the local flaking was more conspicuous with a prolonged oxidation duration. The cross-sectional morphology highlights the augmented thickness of the oxide film on the alloy's surface postoxidation at 1233.15 K. The oxide film thickness measured approximately 3, 4, and 5 μm following oxidation at 100, 300, and 500 h, respectively. During 300 h of oxidation, the granular $Mn$-$Cr$ spinel present in the most superficial layer of the oxide film predominantly flaked off, while the inner $Cr_2O_3$ film remained continuous and intact. However, following oxidation after 500 h, localized cracking in the $Cr_2O_3$ film was also apparent.

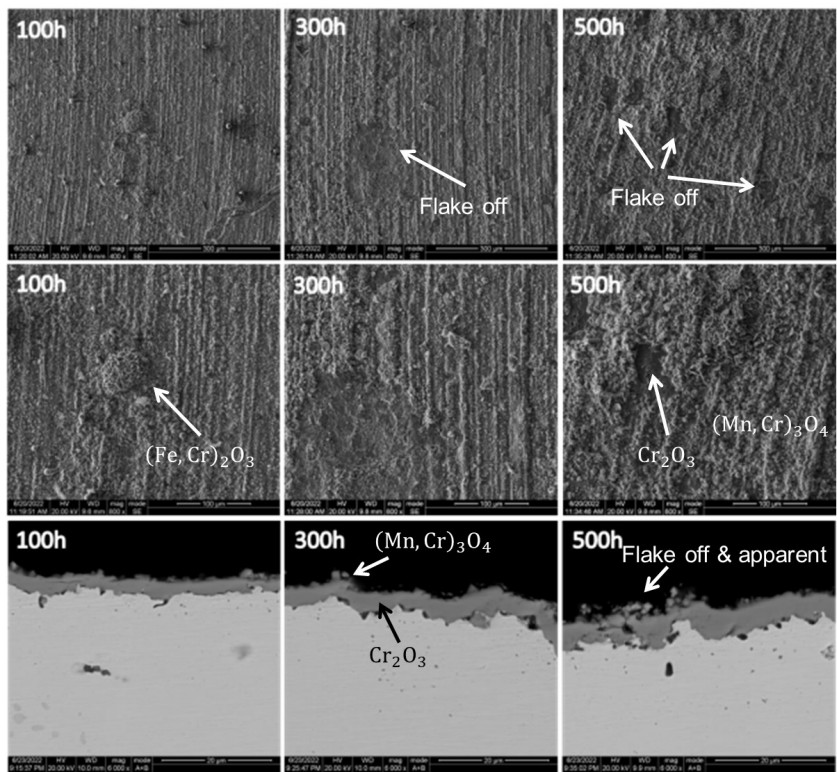

**Figure 9.** The oxidation morphology of 2535 *Nb* alloy sample at 1233.15 K after 100 h, 300 h, and 500 h.

### 3.1.3. The Oxidation Morphology of 2535 *Nb* Alloy Sample at 1323.15 K

The microscopic morphology of the 2535 *Nb* alloy sample subsequent to oxidation at 1323.15 K is displayed in Figure 10.

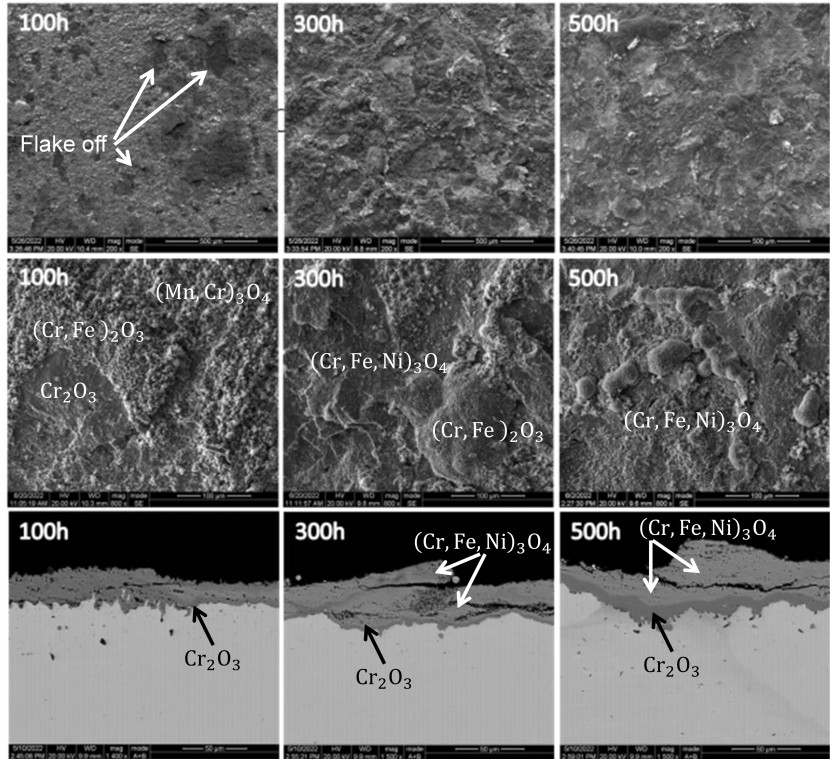

**Figure 10.** The oxidation morphology of 2535 *Nb* alloy sample at 1323.15 K after 100 h, 300 h, and 500 h.

The figure exhibits that the oxidation corrosion of the alloy was relatively severe at this temperature as the surface oxide appeared to exfoliate in large areas in a lamellar manner. The high magnification photograph reveals that the oxide particles in the most superficial layer were significantly shaped, loose, and porous, while the oxide particles in the area after exfoliation were smaller and relatively dense in structure. Upon 300 and 500 h of oxidation, the surface underwent nearly complete exfoliation and regrowth. The cross-sectional morphology demonstrated that the oxide film could be divided into two layers: the inner layer was a uniform and dense dark gray $Cr_2O_3$ layer, while the outer layer was light gray with a relatively loose structure containing many pores, which is an oxide containing $Ni$, $Fe$, and $Cr$. These observations suggest that the corrosion resistance of the alloy declined as the temperature increased to 1323.15 K. In addition to the $Cr_2O_3$ formation, the oxidation of $Ni$ and $Fe$ in the alloy also occurred. The crack expansion direction in the oxide film was roughly parallel to the surface, which is consistent with the lamellar exfoliation structure observed in its surface morphology and indicated the periodic behavior of the alloy surface oxide film exfoliation and regeneration during the oxidation process.

The outcomes of the oxidation tests of the 2535 $Nb$ alloy in the air at 1163.15 K, 1233.15 K, and 1323.15 K demonstrated that at lower temperatures, the alloy surface can form a single, continuous $Cr_2O_3$ film with exceptional oxidation resistance. However, as the temperature increased, the oxidation resistance decreased, and the oxidation of $Fe$ and $Ni$ in the alloy was promoted, which resulted in the formation of a double oxide film with $Cr_2O_3$ as the inner layer and oxides containing $Ni$, $Fe$, and $Cr$ as the outer layer. At a specific temperature, the oxide film continued to grow as the oxidation time increased, and it experienced rupture and flaking. This phenomenon occurred due to the bending and deformation of the oxide film under the growth stress and the formation of folds. When the stress accumulation was substantial and the oxide film could not pass the plastic deformation, it led to rupture and the subsequent formation of a new oxide film. Consequently, the oxide film underwent periodic cracking and peeling, healing, and regeneration. At 1323.15 K, the oxide film growth rate was significant, and the formation and expansion of the cracks in the film were also rapid. The cracks and spalling could not self-heal and regenerate in time, which resulted in obvious cracks in the film. Therefore, the corrosion mechanism gradually changed after the temperature exceeded 1323.15 K, and the research range was chosen within the range of [1073.15,1273.15] in K.

*3.2. Metal Sheets Test*

To establish a more precise quantitative model of corrosion, we processed the 2535 $Nb$ alloy samples into 15 mm ×10 mm × 1.5 mm samples for testing to achieve a weight gain of 0.01 mg.

Prior to testing, the 15 samples underwent necessary pretreatment procedures such as grinding, polishing, cleaning, drying, and initial dimensional and mass measurements, as shown in Figure 11a. To ensure consistency among the samples, we attempted to maintain the same surface roughness of each sample as much as possible during the polishing process.

For the test, we selected five temperature gradients of 1073.15 K, 1123.15 K, 1173.15 K, 1233.15 K, and 1273.15 K (in air), as Figure 11b shows. The samples were weighed every 20 h for a total of 100 h. Figure 12 shows the weight gain data of each sample divided by its surface area.

The corrosion rate of the 2535 $Nb$ alloy at each temperature step could be determined by calculating the average incremental weight of three parallel samples according to Equation (1), as listed in Table 3.

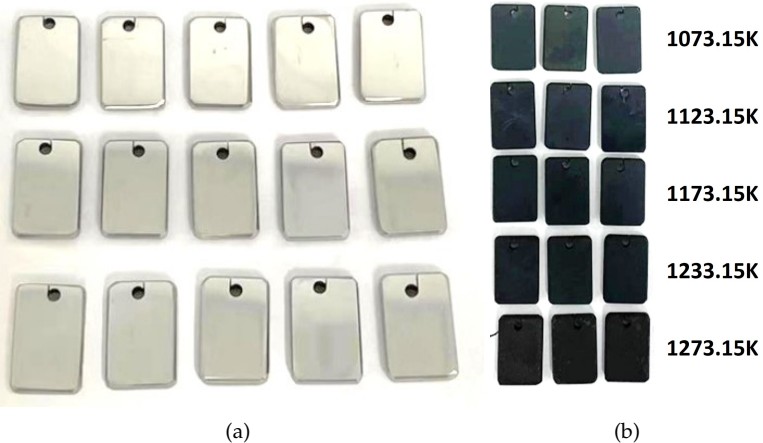

(a)                                                (b)

**Figure 11.** The alloy sheets for test. (**a**) Before test. (**b**) After test.

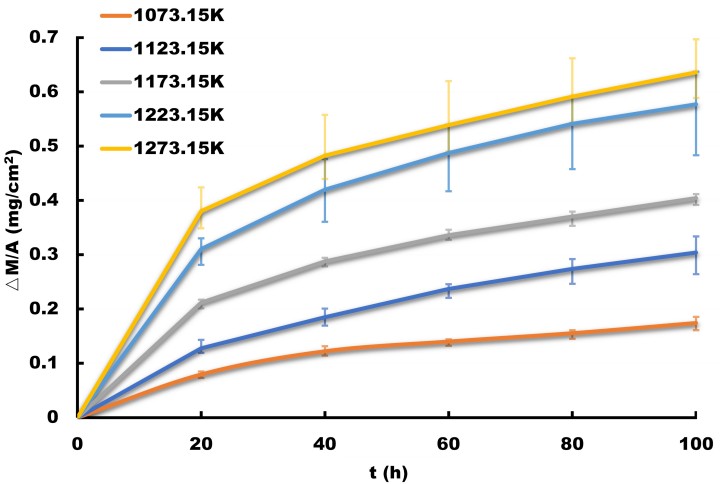

**Figure 12.** The weight gain per unit surface data of sample with time.

**Table 3.** Corrosion rate at each temperature gradient.

| T/K | 1073.15 | 1123.15 | 1173.15 | 1233.15 | 1273.15 |
|---|---|---|---|---|---|
| $k_p/\mathrm{mg}^2\mathrm{cm}^{-4}\mathrm{h}^{-1}$ | $2.99 \times 10^{-4}$ | $9.44 \times 10^{-3}$ | $1.606 \times 10^{-3}$ | $3.309 \times 10^{-3}$ | $3.847 \times 10^{-3}$ |

According to Equation (2), the linear relationship between the corrosion rate and temperature $T$ can be obtained by plotting the corrosion rate data against the corresponding temperature values and fitting a curve to the data points, as shown in Figure 13.

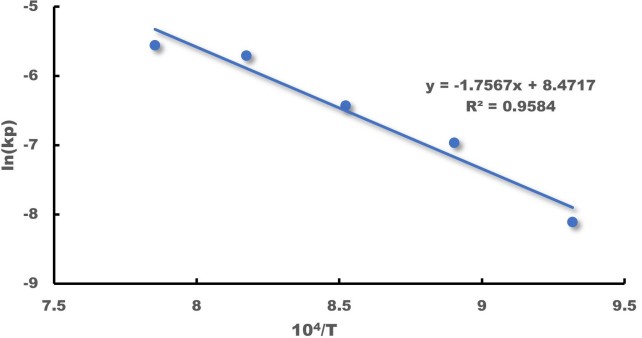

**Figure 13.** The relationship between kp and T in the materials corrosion field.

Given the weight gain threshold as 53.13 mg/cm$^2$, the lifetime of the samples can be calculated as follows

$$\hbar^2 / k_p = t \qquad (10)$$

where $\hbar$ denotes the threshold and t represents the lifetime of the samples. Hence, the failure times of the samples at different temperature steps are shown in Table 4. It is noted that the choice of the weight gain threshold may vary based on engineering needs, but it does not significantly affect the conclusions of this study.

**Table 4.** Pseudo failure lifetime at each temperature gradient.

| T/K | 1073.15 | 1123.15 | 1173.15 | 1233.15 | 1273.15 |
|---|---|---|---|---|---|
| t/h | $7.595 \times 10^6$ | $3.665 \times 10^6$ | $1.882 \times 10^6$ | $1.020 \times 10^6$ | $5.805 \times 10^5$ |

In the field of materials, it is common practice to calculate the lifetime value of the 2535 *Nb* alloy at various temperatures. However, this approach alone is insufficient to fully characterize the reliability of this material in a high-temperature corrosive environment. Obtaining the life distribution of this material in such an environment is necessary for the accurate evaluation of the instrument's life.

### 3.3. Parameter Estimation of Pseudo Failure Lifetime Model

Based on the data obtained in the previous section, three parallel samples were placed on each temperature gradient, and the incremental weights of the samples were arranged from smallest to largest on each temperature gradient, as listed in Table 5.

**Table 5.** Corrosion rate ranking at each temperature gradient.

| T/K | | 1073.15 | 1123.15 | 1173.15 | 1233.15 | 1273.15 |
|---|---|---|---|---|---|---|
| | min | $2.54 \times 10^{-4}$ | $7.25 \times 10^{-4}$ | $1.50 \times 10^{-3}$ | $2.29 \times 10^{-3}$ | $3.29 \times 10^{-3}$ |
| $k_p / \mathrm{mg^2 cm^{-4} h^{-1}}$ | med | $3.18 \times 10^{-4}$ | $1.03 \times 10^{-3}$ | $1.66 \times 10^{-3}$ | $3.75 \times 10^{-3}$ | $3.64 \times 10^{-3}$ |
| | max | $3.29 \times 10^{-4}$ | $1.10 \times 10^{-3}$ | $1.66 \times 10^{-3}$ | $4.02 \times 10^{-3}$ | $4.68 \times 10^{-3}$ |

To obtain three quantitative models between the corrosion rate and temperature *T*, the minimum, medium, and maximum weight gain values for each temperature step were modeled and calculated, as listed in Table 6. The fitting deviation R2 of all three curves was found to be not less than 90%, as shown in Figure 14.

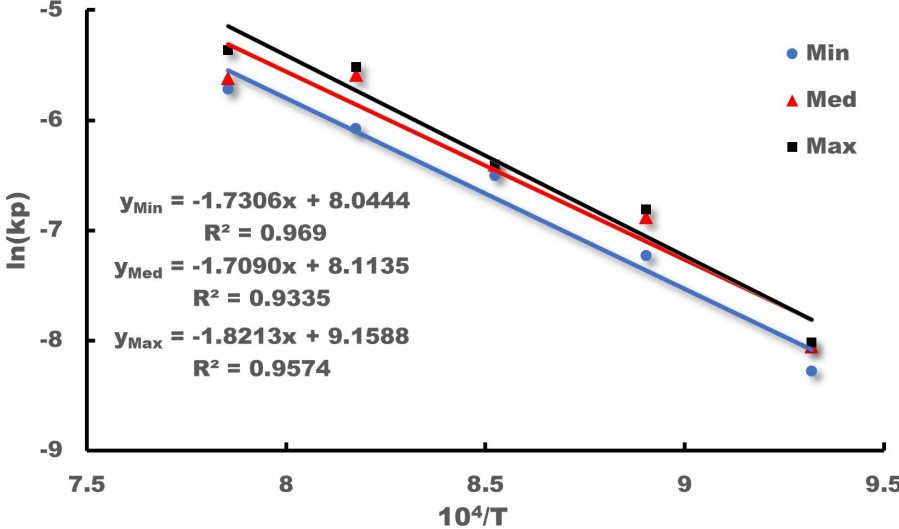

**Figure 14.** Data fitting for the pseudo lifetime model.

**Table 6.** Crucial parameters of corrosion model at each temperature gradient.

| T/K | 1073.15 | 1123.15 | 1173.15 | 1233.15 | 1273.15 | Slope b | Inter a |
|---|---|---|---|---|---|---|---|
| $x = 1/T \times 10^3$ | 9.318 | 8.904 | 8.524 | 8.176 | 7.854 | / | / |
| $y_{min} = ln(\eta)$ | −8.279 | −7.229 | −6.502 | −6.079 | −5.718 | −1.731 | 8.044 |
| $y_{med} = ln(\eta)$ | −8.054 | −6.881 | −6.401 | −5.585 | −5.615 | −1.709 | 8.113 |
| $y_{max} = ln(\eta)$ | −8.019 | −6.812 | −6.400 | −5.516 | −5.365 | −1.821 | 9.159 |

The pseudo failure lifetime of each of the 15 groups of samples is calculated separately, as shown in Table 7. The unit of lifetime is converted to year for easy observation and is represented by "a".

**Table 7.** Pseudo lifetime ranking at each temperature gradient.

| | T/K | 1073.15 | 1123.15 | 1173.15 | 1233.15 | 1273.15 |
|---|---|---|---|---|---|---|
| | min | 1042.58 | 508.56 | 263.70 | 144.29 | 82.77 |
| Pseudo lifetime/a | med | 795.32 | 391.44 | 204.68 | 112.82 | 65.17 |
| | max | 795.89 | 373.86 | 187.33 | 99.32 | 55.34 |

The exponential, Weibull, and log–normal distributions are introduced to fit the pseudo lifetime of the samples. According to Equation (6), the log-likelihood function for each distribution is established as follows

$$\begin{cases} \ln(L_{Exp}) = \sum\limits_{i=1}^{k} n \ln\left[\lambda e^{-\lambda t_i}\right] = \sum\limits_{i=1}^{k} n_i \ln\left[\frac{e^{-(t_i/Be^{C/T_i})}}{Be^{C/T_i}}\right], t_i \sim Exp(\lambda) \\ \ln(L_W) = \sum\limits_{i=1}^{k} n \ln\left[\frac{\beta}{B \cdot e^{C/T_i}}\left(\frac{t_i}{B \cdot e^{C/T_i}}\right)^{\beta-1} e^{-\left(t_i/B \cdot e^{C/T_i}\right)^{\beta}}\right], t_i \sim W(\beta, \eta) \\ \ln(L_{LN}) = \sum\limits_{i=1}^{k} n \ln\left[\frac{1}{\sigma t_i}\phi\left(\frac{\ln(t_i) - \ln(B) - C/T_i}{\sigma}\right)\right], t_i \sim LN(\mu, \sigma^2) \end{cases} \quad (11)$$

in which $L_{EXP}$, $L_W$, and $L_{LN}$ indicate the likelihood function of the exponential, Weibull, and log–normal distribution, respectively. $k$ is the number of stress levels, and $n$ is the number of samples at each level. With the maximum likelihood estimation method, the parameters of each distribution can be obtained: $\lambda = 877.69$, $B_{Exp} = 7.27 \times 10^{-5}$, and $C_{Exp} = 17498.95$ for the exponential distribution; $\beta = 7.12$, $\eta = 925.65$, $B_W = 8.85 \times 10^{-5}$, and $C_W = 17345.90$ for the Weibull distribution; and $\mu = 6.769$, $\sigma = 0.146$, $B_{LN} = 6.97 \times 10^{-5}$, and $C_{LN} = 17536.07$ for the log–normal distribution. It should be noted that the values of $\lambda$, $\beta$, and $\mu$ are taken at the stress level of T = 1073.15 K.

By substituting the parameters and data in Table 7 into Equation (11), we have $L_{Exp} = −96.628$, $L_W = −73.914$, and $L_{LN} = −75.115$. Therefore, the Weibull distribution model with the largest value is selected as the best-fit distribution. Additionally, it is worth noting that the value of $L_{LN}$ is only slightly smaller than $L_W$, which implies that for the existing data, a log–normal distribution model is also available as an alternative.

Consequently, the reliability function of the material at different stress level is established based on Equation (7), and the lifetime of the samples at each level with varying values of reliability can be obtained, such as R = 90%, R = 70%, and R = 36.8% (the characteristic life), as shown in Table 8 and Figure 15. In addition, $\eta_{Tra}$ denotes the transformed lifetime data in Table 4 and $R_{Tra}$ denotes the reliability of the traditional corrosion algorithm at the pseudo lifetime, which reveals that the conventional method used in the field of material has a reliability of approximately 50%.

**Table 8.** Comparison of lifetime at each temperature gradient in proposed and traditional models.

| T/K | 1073.15 | 1123.15 | 1173.15 | 1233.15 | 1273.15 |
|---|---|---|---|---|---|
| $\eta_{0.9}$/a | 674.78 | 328.59 | 209.20 | 92.96 | 53.26 |
| $\eta_{0.7}$/a | 800.85 | 389.99 | 239.55 | 110.33 | 63.21 |
| $\eta_{0.368}$/a | 925.60 | 450.74 | 262.40 | 127.52 | 73.06 |
| $\eta_{Tra}$/a | 867.01 | 418.38 | 214.84 | 116.44 | 66.27 |
| $R_{Tra}$/% | 51.15 | 53.74 | 54.28 | 54.770 | 55.23 |

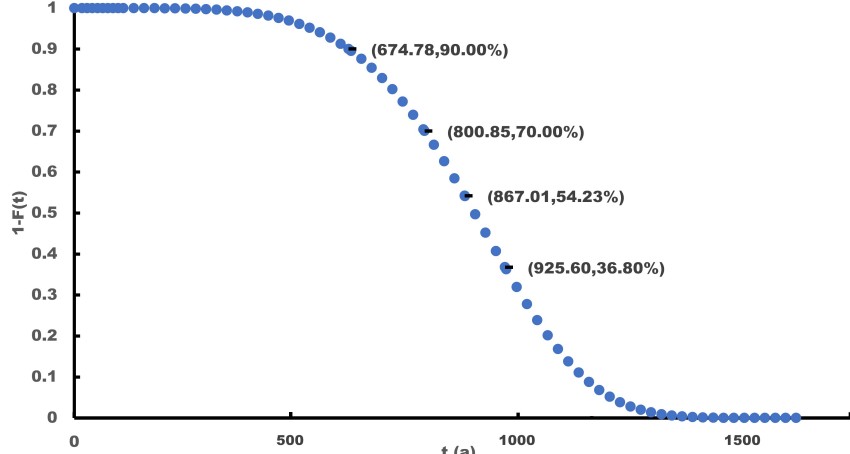

**Figure 15.** Schematic of the reliability function $R(t)$ for pseudo lifetime model at 1073.15 K.

### 3.4. Acceleration Factor

The estimate of the activation energy $Ea = 1.495\,eV$ can be obtained by substituting $C_W$ and $k_B$ into $C_W = E_a/k_B$. Additionally, the acceleration factors for the other temperature conditions can be calculated by using 1073.15 K as the reference temperature, as shown in Table 9.

**Table 9.** Acceleration factors for different temperature gradients.

| T/K | 1073.15 | 1123.15 | 1173.15 | 1233.15 | 1273.15 |
|---|---|---|---|---|---|
| AF | 1 | 2.065 | 4.009 | 7.371 | 12.922 |

Hence, we can utilize temperature as the accelerating stress and employ the acceleration method to efficiently evaluate the corrosion life of the 2535 *Nb* alloy. For instance, if the temperature stress is raised to 1273.15 K, the acceleration multiplier would be approximately 13 times larger.

## 4. Conclusions

This study focused on investigating the high-temperature oxidative corrosion characteristics of a 2535 *Nb* alloy, and the results showed that it exhibited good oxidative corrosion resistance within the range of $[1073.15, 1323.15]$ in K. To establish a degradation model for the 2535 *Nb* alloy under high-temperature oxidation corrosion, the pseudo failure life analysis method was utilized, which differs from the general algorithm for corrosion of metallic materials. The model developed can provide the corrosion life distribution of metallic materials. The reliability of the corrosion life values given by the general algorithm for metallic materials was found to be close to 50% in this study. The shape factor $\beta = 4.989$ of the corrosion life distribution of the 2535 *Nb* alloy fitted with the Weibull function had a shape that was similar to a normal distribution. Based on this model, an accelerated life test model can be easily derived with temperature as the acceleration factor. The corrosion rate of the 2535 *Nb* alloy in a 1273.15 K environment was found to be almost 13 times higher

than that in a 1073.15K environment. It is worth noting that the oxide film generated by the 2535 *Nb* alloy in a 1323.15 K environment exhibited obvious cracks, which implies that the high-temperature oxidation corrosion model of the 2535 *Nb* alloy may not be suitable for working in environments exceeding 1323.15 K.

This study employed laboratory life test data from material-level samples to develop an accelerated life model. Future research will involve scholars conducting an accelerated degradation model investigation to explore the degradation of the operational parameters of the whole thermocouple instrument under operating conditions. This will improve the life assessment findings of this paper to account for the effects of the actual usage environment and individual disparities on laboratory test outcomes.

**Author Contributions:** Conceptualization: P.L.; methodology: H.Z. and P.L.; validation: S.L. and Z.Y.; investigation: H.Z. and P.L.; data curation: S.L.; writing—original draft preparation: H.Z.; writing—review and editing: P.L.; supervision: Y.L. All authors have read and agreed to the published version of the manuscript.

**Funding:** The Science and Technology Program of Guangzhou, China (No. 202201010303).

**Institutional Review Board Statement:** Not applicable.

**Informed Consent Statement:** Not applicable.

**Data Availability Statement:** The data for this study are available upon request from the corresponding author.

**Conflicts of Interest:** The authors declare no conflict of interest.

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
