# Peer review of "An Accelerated-Based Evaluation Method for Corrosion Lifetime of Materials Considering High-Temperature Oxidation Corrosion"

_sustainability, doi:10.3390/su15119102_

Round 1

Reviewer 1 Report

Reviewer’s Comments on sustainability-2390383

An accelerated-based evaluation method for corrosion lifetime of materials considering high temperature oxidation corrosion

This paper presents a study about metal corrosion resistance assessment. An accelerated life evaluation model is presented based on pseudo failure times that are fitted to a Weibull distribution. The authors provided the reliability curves and accelerating factors for an 535Nb alloy. I have the following comments:

1.     I believe that the authors need to provide and discuss more relevant published works in the introduction section that are related to reliability modeling of the considered case study. The authors need to discuss more details about these works in the aim of highlighting the contributions of the manuscript.

2.     Degradation modeling is an important research area that needs to be more deeply discussed in the manuscript. Several models have been proposed in the literature such as stochastic models. The authors do not discuss this area enough, I believe this is an important topic as it is directly related to the case study.

3.     Section 2.3 is titled “Acceleration factor model based on the proportional hazard model”, but the proportional hazard rate models is not presented and discussed. Instead, the authors present a linearized cumulative function of the Weibull distribution and the accelerating factor.

4.     As the Weibull distribution is considered for the pseudo failure times, the authors are encouraged to provide goodness of fit tests to verify if the Weibull distribution is the best probability distribution function for the data.

5.     Why did the authors consider a non-parametric approximation (median ranks) for the estimation of the parameters of the Weibull distribution. I would encourage the authors to estimate the parameters with a parametric method, such as maximum likelihood estimation method.

6.     In section 3.4, the authors mention “By substituting Equation 9, the activation energy Ea can be calculated”, but when it is substituted where?, in the fitted linear models? Which ones? Please provide more details in the manuscript.

7.     The authors are encouraged to be more specific about the figures and tables titles, most are quite general, please declare which variables are considered and which data is used to construct the illustrations.

8.     What is difference in considering the temperature in Kelvin and Celsius, as the Arrhenius relationship is considered?  Kelvin degrees as most frequently used in the literature.

9.     Apart from the R^2 obtained in the linear models in Figure 11, the authors are encouraged to consider other models, such as power, logarithmic and exponential, then these models can be compared to select the one that best describes the trajectories.

10. The authors are encouraged to provide insights for future research in the conclusion section.

No comments.

Author Response

Thank you for your comments on our manuscript entitled " An accelerated-based evaluation method for corrosion lifetime of materials considering high temperature oxidation corrosion" (Manuscript Number: sustainability-2390383). Those comments are very helpful for revising and improving our paper, as well as the important guiding significance to our researches. We have studied the comments carefully and made corrections which we hope meet with approval. The responces are in the attachment. 

Reviewer 2 Report

Review Report

Article title: An accelerated-based evaluation method for corrosion lifetime of materials considering high temperature oxidation corrosion

The manuscript presents combines the standard approach for evaluating metal corrosion resistance in the field of materials with the method for assessing component life in the domain of reliability. The study describes a thermocouple accelerated life evaluation model that enhances the accuracy and efficiency of life evaluation for related products.

Specific comments:

1. The subchapter 2.1 presents a well-known theory that can be omitted. Furthermore, in the subsection 2.1, it would be appropriate to use the Ellingham diagram instead of Table 1.

2.     Under each equation, all symbols should be explained.

3.     The authors found Cr2O3 as the main oxidation product (Figs. 6, 7, 8). However, there is no Cr in the alloy as indicated in Table 2. Shouldn't there be Cr instead of Ge in table 2?

4.     What was the annealing atmosphere used during the oxidation experiments? Air?

5.  The temperature 950 °C should be indicated under Fig. 8. Under Fig. 9, temperature 1050 °C should be written instead of 890 °C.

6.     In the subsections 3.1.1 – 3.1.3 it is necessary to provide EDS results to confirm the elements. Furthermore, an XRD should be presented to confirm the phases.

7.    A specific mass gain should be given in weight gain table. The mass must be divided by surface area. Data in Table 3 should be processed statistically and presented graphically. The average and standard deviation of the measurements should be calculated and plotted as a function of time.

8.     What is the dimension of the physical quantity kp in Table 4?

9.  In Table 6 a specific mass gain should be listed instead of mass gain. Furthermore, data should be presented as an average ± standard deviation. Also Fig. 11 should include an average specific mass gain plus standard deviation instead of individual points.

10.  In Table 9, it would be better to convert the time values to years instead of hours.

11.  The novelty of the work is quite weak since only a lifetime prediction model was tested.

12.  In the sentence in line 300, the dimension of activation energy is not mentioned.

13.  Activation energies should also be determined from Figs. 10, 11.

Author Response

(The authors gave the same response as above.)

Reviewer 3 Report

1. In the introduction section, in addition to introducing stainless steel, nickel-based high-temperature alloys, iron-based corrosion resistant alloys and precious metal materials, it is also appropriate to introduce the metal copper. Copper is a good conductive metal with good thermoelectric properties, good corrosion resistance and workability, making it one of the ideal materials for thermocouples. The relevant literature should be cited "Green Energy Environ. , DOI: 10.1016/j.gee.2022.01.005". 

2. The aesthetics of the figures need to be improved and the titles of Figures 6, 7 and 8 need to be revised. The formatting of all tables should be consistent, and whether graphing the data in Table 3 would be clearer.

The paper's grammar and academic expression are rough. It is recommended to further enhance and polish the language, otherwise, it will be difficult to attract readers.

Author Response

(The authors gave the same response as above.)

Round 2

Reviewer 1 Report

The authors have considered the provided comments to improve the manuscript, I have no further comments.

English is fine

Reviewer 2 Report

Dear Authors,

I appreciate the great efforts you have made in response to my previous questions and comments. You have significantly improved the clarity of your writing and addressed most of my concerns. 

Kind regards,

The Reviewer